# The Increasing Role of CT-Guided Cryoablation for the Treatment of Liver Cancer: A Single-Center Report

**DOI:** 10.3390/cancers14123018

**Published:** 2022-06-19

**Authors:** Claudio Pusceddu, Luigi Mascia, Chiara Ninniri, Nicola Ballicu, Stefano Zedda, Luca Melis, Giulia Deiana, Alberto Porcu, Alessandro Fancellu

**Affiliations:** 1Department of Oncological and Interventional Radiology, Oncological Hospital A, Businco, I-09121 Cagliari, Italy; clapusceddu@gmail.com (C.P.); nicola.ballicu@aob.it (N.B.); stefanozedda1@gmail.com (S.Z.); doclucamelis@tiscali.it (L.M.); 2Department of Medical Oncology, Oncological Hospital A, Businco, I-09121 Cagliari, Italy; l.mascia@aol.com; 3Unit of General Surgery 2, Clinica Chirurgica, Department of Medical, Surgical, and Experimental Sciences, University of Sassari, I-07100 Sassari, Italy; ninnirimariachiara@hotmail.it (C.N.); giulia.deiana2@gmail.com (G.D.); alberto@uniss.it (A.P.)

**Keywords:** cryoablation, liver, hepatocellular carcinoma, HCC, metastases

## Abstract

**Simple Summary:**

Image-guided percutaneous ablation of primary and metastatic liver tumors has been gaining importance in patients who are not suitable for hepatic resection or liver transplantation. The aim of our retrospective study was to assess the effectiveness of CT-guided cryoablation in 49 patients with hepatocellular carcinoma or liver metastases. Our results highlighted that cryoablation is an effective and safe method for the treatment of liver cancers that are not amenable to surgical resection. Furthermore, cryoablation has important advantages when compared to other ablation techniques such as radiofrequency or microwave ablation; therefore, this method should be included in the armamentarium of treatment options in centers dedicated to the multidisciplinary treatment of liver cancer.

**Abstract:**

Purpose: Cryoablation (CrA) is a minimally invasive treatment that can be used in primary and metastatic liver cancer. The purpose of this study was to assess the effectiveness of CrA in patients with hepatocellular carcinoma (HCC) and liver metastases. Methods: We retrospectively evaluated the patients who had CrA for HCC or liver metastases between 2015 and 2020. Technical success, complete ablation, CrA-related complications, local tumor progression, local recurrences, and distant metastases were evaluated in the study population. In patients with HCC, the median survival was also estimated. Results: Sixty-four liver tumors in 49 patients were treated with CrA (50 metastases and 14 HCC). The mean tumor diameter was 2.15 cm. The mean follow-up was 19.8 months. Technical success was achieved in the whole study population. Complete tumor ablation was observed after one month in 92% of lesions treated with CrA (79% and 96% in the HCC Group and metastases Group, respectively, *p* < 0.001). Local tumor progression occurred in 12.5 of lesions, with no difference between the study groups (*p* = 0.105). Sixteen patients (33%) developed local recurrence (45% and 29% in the HCC Group and metastases Group, respectively, *p* = 0.477). Seven patients (14%) developed distant metastases in the follow-up period. Ten patients (20.8%) underwent redo CrA for local recurrence or incomplete tumor ablation. Minor complications were observed in 14% of patients. In patients with HCC, the median survival was 22 months. Conclusions: CrA can be safely used for treatment of HCC and liver metastases not amenable of surgical resection. Further studies are necessary to better define the role of CrA in the multidisciplinary treatment of liver malignancies.

## 1. Introduction

The liver is a common site of primary and metastatic tumors. Primary liver cancer is the sixth most commonly diagnosed cancer, with approximately 906,000 new cases, and the third leading cause of cancer death worldwide, with 830,000 deaths in the year 2020 [1]. Hepatocellular carcinoma (HCC) accounts for 75–85% of primary liver cancer. In Western countries, HCC develops in the presence of an underlying liver disease in the majority of patients. Surgical resection is the most effective treatment for HCC but, unfortunately, the resection rate is less than 30% [2,3]. Liver transplantation is another surgical option in selected patients, but it is limited because of donor shortage and severe complications that can arise from the administration of systemic immunosuppressants [2].

Liver metastases represent a pivotal cause of death in patients with cancer, especially form colorectal cancer, which affects more than 1.3 million people worldwide annually [4]. It is known that 50–60% of patients diagnosed with colorectal cancer develop liver metastases, and 80–90% of these patients have unresectable metastatic liver disease [5].

Currently, patients with HCC and liver metastases can count on many available treatment options, which should be carefully evaluated by an experienced multidisciplinary team. In patients with unresectable tumors, locoregional therapies are directed toward inducing selective tumor necrosis, and include ablation methods, arterially directed therapies, and radiation therapy (RT).

Percutaneous tumor ablation under radiological guidance has been gaining popularity for the management of HCC and metastases, especially in patients not suitable for surgery. Radiofrequency ablation (RFA) is the most clinically verified and used ablation modality, although it has specific limitations. Alternate energy-based technologies using thermal (hot or cold) effect for focal tumor ablation have been recently introduced in clinical use, namely microwave ablation (MWA), cryoablation (CrA), high-intensity focused ultrasound (HIFU), laser therapy, and irreversible electroporation (IRE) [4,6,7,8,9,10,11,12].

The different ablation techniques vary notably in their mode of action, and each of them has benefits and limitations. CrA causes cancer cells’ death using extreme cold through a gas-containing needle inserted percutaneously under image guidance [12,13,14,15]. CrA involves a freeze–thaw–freeze cycle to achieve tumor necrosis. Cellular necrosis and death occur via a direct mechanism, which causes cold-direct cold injury, as well as via an indirect mechanism, which causes modifications of the cellular microenvironment and impaired tissue viability [6,13,16,17].

CrA is considered a safe alternative to RFA or MWA in patients with liver malignancy where these two methods are contraindicated or not advisable based on tumor position.

CrA presents some important benefits when compared to more employed ablation techniques, as it has the ability to monitor through imaging the ice-ball formation during the procedure. Although CrA has been used for tumor ablation since the 1990s, reports on the use of CrA for liver tumors are scarce in the current literature; in particular, only a few articles come from European institutions, where CrA is used less than in the US and Asian countries.

The aim of this study is to investigate the safety and effectiveness of CrA in the management of HCC and liver metastases, by reporting on a recent case series from a European center with expertise in various image-guided ablation methods.

## 2. Materials and Methods

### 2.1. Study Population

In the last 10 years, RFA, CrA, and MWA have been used at our institution for thermal ablation of primary and metastatic malignancies [12,13]. We have started using CrA for patients with liver cancer in 2015. Between 2015 and 2020, a total of 1092 image-guided percutaneous ablations (including RFA, MWA, and CrA) were carried out for lung, breast, liver, kidney, and osseous tumoral lesions. Sixty-four CrA, 15 RFA, and 73 MWA were performed for liver malignancies. For the purposes of this study, an institutional review was conducted on patients with hepatic tumors who received CrA between 2015 and 2020. Patients with one or two HCC lesions, or one or two hepatic metastases, were submitted to CrA with the aim of obtaining complete tumor ablation. None of the patients included in the present study was considered suitable for liver resection or transplantation for one or more of the following reasons: location and number of liver lesions, presence of distant metastases, preexisting comorbidities, prediction of insufficient liver remnant, or patient’s decision. Absence of distant metastases, a Child-Pugh class A or B, a platelet count > 100,000/mm^3^, and a prothrombine time > 65% were required to receive liver CrA. The following data were extrapolated for the study population: age, sex, and tumor characteristics (size, number, localization of lesions, type liver cancer [HCC or metastases]). The operation notes were used to report the technique of CrA procedures, and procedure-related complications. Follow-up data included the incidence and time to local recurrence and death. The study was approved by the institutional review board.

### 2.2. Preoperative Evaluation and CrA Procedure

Liver ultrasound, 3-phase contrast enhanced liver CT, and total body CT scan were performed preoperatively in all patients. Each case was discussed during a multi-disciplinary tumor board that included hepato-biliary surgeons, medical oncologists, hepatologists, and interventional radiologists. Patients with HCC or liver metastases measuring less than 3 cm in major diameter were usually treated with RFA, whereas tumors larger than 3 cm were primarily approached with MWA, although we have also gradually replaced RFA with MWA for the ablation of small tumor lesions. The decision to resort to CrA instead of heat-based methods was mainly conditioned by the proximity of the lesions to important anatomical structures (diaphragm, gallbladder, major biliary and vascular structures, chest wall, bowel, and heart).

All percutaneous CrA procedures were performed using an argon-based CrA unit (Visual Ice Cryoablation System, Boston Scientific Corporation, Marlborough, MA, USA). The system comprises a computer workstation, a gas distribution apparatus (argon and helium), and 17-gauge cryoprobes. Once the needles were inserted into the tumor, argon gas was circulated inside the needles to cause rapid freezing of surrounding tissues and formation of an iceball. After 10 min of tumor freezing, a rapid active thawing of the ice sphere was performed, by using the same needles to circulate helium gas (4 min), followed by slow passive thawing (4 min). The freeze-thaw cycle was repeated twice.

The iceball effect on the tissues can be regulated depending upon the distance from the needle. For example, temperatures ranging between 20 and 40 degrees below zero Celsius result in intracellular ice formation with membrane rupture and cell death. A delayed vascular injury follows vascular thrombosis causing ischemia and increased cellular death. With temperatures above 20 degrees below zero Celsius, supercooling of tissues occurs without intracellular ice formation. Under these conditions, apoptosis is a possible but erratic mechanism of cellular death, and repeated CrA cycles are sometimes necessary to increase the lethal volume in the peripheral part of the iceball. Importantly, the size of the iceball can be regulated by the characteristics of the cryoprobes employed. With careful positioning and power regulation of individual cryoprobes, multiple coalescing iceballs can be “sculpted” to conglobate tumors with irregular contours. In selected cases, it is also possible to move organs away from the surface of the iceball by introducing air or fluid around them.

For CrA treatment, we used two types of 17G needles: IceRod PLUS and IceSphere (Boston Scientific International B.V, Vestastraat 6, EX Kerkrade, The Netherlands). IceRod PLUS probes permit an ellipsoid iceball formation of 16 mm × 40 mm at −40 °C, whereas IceSphere probes cause and iceball formation of 15 mm × 23 mm at −40 °C. All the procedures were carried out under CT guidance (CT device with 5-mm collimation at 80–140 mA SOMATOM Sensation, Siemens, Forcheim, Germany).

All CrA procedures have been performed under local anesthesia and conscious sedation, with the patient in a supine, prone, or lateral decubitus position, depending on the location of the lesion in order to choose the best access while avoiding sensitive structures, and to reduce potential complications. An anesthesiologist performed the conscious sedation and monitored the patient during the procedure and the postoperative period.

Usually, patients were discharged the day after the CrA procedure.

Explicative cases of CrA procedure were reported in Figure 1 and Figure 2.

### 2.3. Definitions and Follow-Up

The technical success of CrA was considered when the tumor lesion was completely included in the iceball at the end of the procedure. Efficacy of CrA was evaluated in terms of complete response with CT scan and contrast-enhanced ultrasound at 1, 3, 6, and 12 months for the first year after treatment and every 6 months thereafter. MRI and PET-CT scans were used in selected cases. Ablation was considered complete when no contrast enhancement was observed in the arterial phase at the 1-month CT-scan. Ablation was considered incomplete when a residual contrast enhancement (linear or nodular) was observed. The size of the CrA zone at the 1-month CT scan was used as the basal assessment to which subsequent follow-up imaging was compared. Increase in the diameter of the lesion treated was interpreted as local tumor progression. Steadiness or decrease in the diameter of the ablation area was considered as successful ablation. Local recurrence was defined as the appearance of new tumor lesions in the remnant liver.

### 2.4. Study Endpoints and Statistical Analysis

Technical success, complete ablation, local tumor progression, and local tumor recurrence were the study endpoints. Continuous variables were presented as mean ± SD. Categorical variables were compared by means of a Chi-square test or Fisher exact text as appropriate, and continuous variables were assessed by the *t*-test or the Mann–Whitney test. *p*-values < 0.005 were considered statistically significant. Technical success, complete response, and local tumor progression were calculated by tumor type (HCC or metastases) and compared with Pearson’s chi-square test. In patients with HCC, mortality and median survival were also calculated. Analysis was conducted using IBM SPSS Statistics version 20 (IBM Corporation 2011).

MWA or RFA would have resulted in a more painful procedure because of the proximity of the tumor lesion to the Glisson sheath. The limits of the iceball visible in (c) correspond to the limits of the area of necrosis observed at the 1-month CT scan (d). Intraprocedural monitoring of the ablation area with MWA or RFA is less accurate than with CrA.

## 3. Results

### 3.1. Demographic and Operative Outcomes

Forty-nine patients, 23 men and 26 women, met the inclusion criteria and constituted the study population. The mean age was 67.5 years (range 44–85 years). In total, 64 CrA procedures for 54 tumor lesions were carried out (50 metastases and 14 HCC). The mean tumor diameter was 2.15 cm (range 0.5–5 cm). The metastases were from colorectal (n = 23), breast (n = 12), pancreatic (n = 7), lung (n = 3), thyroid (n = 2), gastric (n = 1), ovarian (n = 1), and cervical cancer (n = 2). Fifty-four patients (84%) received CrA for one tumor lesion, and five (16%) for two tumor lesions. CrA-treated tumors were located in the right liver in 24 cases (37.5%), and in the left liver in 40 (62.5%) (Table 1). Mean follow-up was 19.8 months (range 1–60 months). Technical success was achieved in all the study population. Complete tumor ablation was observed in 92% of lesions treated with CrA (79% in the HCC group and 96% in the metastases group, *p* = 0.032). Sixteen patients (33%) developed local recurrence, 5 (45%) and 11 (29%) in the HCC group and metastases group, respectively (*p* = 0.477). Globally, 20 local recurrences were registered in the study population (6 in the HCC group and 14 in the metastases group; *p* = 0.289). Local tumor progression occurred in one patient (7%) with HCC and in five (10%) with metastases (*p* = 0.105). Seven patients (14%) developed distant metastases during the follow-up. Ten patients (20%) underwent redo CrA for local recurrence or incomplete tumor ablation (Table 2).

Minor complications occurred in seven patients (14%). Two patients (4%) developed a generalized sensation of freezing, albeit in presence of normal body temperature. In both cases, the symptoms resumed in one hour with coverage with a warm blanket and saline infusion. Moderate pain requiring NSAIDs administration was observed in two patients (4%). In two cases (4%), self-limiting liver bleeding occurred. No abscess formation, biliary leaks, bilomas, or hematologic changes were noted.

The mean duration of the CrA procedure was 67.3 ± 9.5 min (range 55 to 77 min), which in our experience is longer than RFA (50–65 min) and MWA (45–60 min).

### 3.2. Subgroup of Patients with HCC

Eleven patients received CrA for HCC with a mean size of 19 mm. In total, 14 HCC lesions were cryoablated. Complete tumor ablation was obtained in 11 HCCs (79%). Five patients experienced a local tumor recurrence, and one patient developed distant metastases. After a median follow-up of 17 months, six patients died from disease progression. The median survival in the HCC group was 22 months. As for operative outcomes by tumor type (HCC or metastases), there was no difference in rates of local recurrence in the liver remnant (*p* = 0.289), local tumor progression (*p* = 0.105), and distant metastases (*p* = 0.607), whereas percentage of complete tumor ablation was significantly higher in the metastases group (*p* < 0.05) (Table 2).

## 4. Discussion

Recent advances in perioperative care and improvements in multidisciplinary treatment of liver tumors has expanded the indications to minor and major hepatectomy [3,18]. Nonetheless, only 5–25% of patients with HCC and 20–25% of those with liver-only metastases are candidates for hepatic resection or transplantation [6,9,19]. CrA has been applied for years in the treatment of both primary and metastatic liver tumors. However, it is less commonly used than other percutaneous ablative methods, such as RFA or MWA.

The results of this study highlighted that CrA is an effective and safe method for the management of selected patients with liver tumors. Among locoregional therapies directed toward inducing selective tumor necrosis, percutaneous heat-based ablation techniques and cryoablation represent efficacious and minimal methods that have been increasingly used in patients with liver cancer. Among them, RFA has been the most employed method; thus, robust evidence exists on the safety and efficacy of that technique in the treatment of metastases and HCC. Studies comparing RFA to resection in patients with HCC have shown that resection is generally associated with better survival outcomes than RFA, although surgery is associated with more complications and morbidity [18,20]. Nonetheless, some studies have underscored that RFA might be considered the first-line treatment in selected patients with HCC lesions ≤2 cm, provided that they are in an accessible location and away from major vascular and biliary structures and adjacent organs. Peng et al. reported a 5-year OS of 80% in patients with central HCC tumors ≤2 cm treated with RFA [21].

MWA has also been shown to be useful in selected patients with either unifocal or multifocal liver tumors [6,8,10,11,12], although data remain relatively limited. In the only randomized controlled study that compared RFA with percutaneous MWA, no significant differences were found between these two techniques in terms of therapeutic effectiveness, complications, and the rates of residual foci of untreated disease [22]. In a randomized study that evaluated the efficacy of MWA versus surgical resection in the treatment of HCC, MWA was associated with lower disease-free survival rates with no differences in OS rates [23]. In a single institution report, we evaluated the use of MWA in 32 patients with 45 HCC tumors. Complete ablation was obtained in 94.1% of small tumors (<3 cm). One-year OS was 82.7%, 2-year survival was 68.9%, and 3-year survival was 55.2% [12].

In general, MWA is associated with lower operative times than RFA, and also permits the simultaneous use of multiple antennas, thus resulting in a more versatile method. For these reasons, at our center, MWA has largely replaced RFA in the percutaneous treatment of HCC and liver metastases, in line with current trends in many interventional radiology centers [24].

Although RFA and MWA have shown their usefulness in liver ablation, they are generally considered not advisable when the lesion to ablate is adjacent to critical structures [4,6]. In fact, CrA is preferable for treatment of lesions located in close vicinity to important anatomical structures, due to the lower risk of tissue damage beyond the area of tumor ablation necrosis [9,11]. In line with current trends, we have carried out CrA in cases where the tumor mass was adjacent to critical structures. In addition, the heat sink effect observed with RFA and MWA may lead to ineffective ablation of perivascular tumors, an effect that does not occur with CrA [19].

In our series, all CrA procedures have been performed under local anesthesia and conscious sedation. It is recognized that the pain related to CrA is lower than other percutaneous ablation techniques [2,4,9,25,26,27]. We have observed this finding mostly in patients with tumor lesions in proximity to the Glisson’s capsule, where the level of sedation and use of opiates were lower than necessary in RFA or MWA.

To note, minor complications occurred in 14% of patients described in the present series. Among them, we have observed in two cases a generalized sensation of freezing immediately after the end of the CrA session, lasting for a couple of hours. In our literature search, we have not found a similar clinical picture, which should be considered different for the well-known cryoshock, a life-threating adverse effect associated to CrA of large liver masses [24,27]. To minimize the risk of that complication, in the present study, no more than two tumoral lesions (with a maximum diameter of 5 cm each) were treated with CrA in a single session, although larger tumors were treated by others [6,17]. Glazer et al., in a cohort of 186 patients submitted to CrA for hepatic tumors, found that tumors smaller than 4 cm were more likely to be treated successfully and without an adverse event [6]. Two patients experienced self-limiting bleeding in our series. It should be noted that CrA lacks an electrocautery needle tract, which bears the risk for CrA-related bleeding, in contrast to RFA and MWA [9,11,25]. Catastrophic hemorrhage and cryoshock reported during the first experiences with CrA in the 1990s have limited the diffusion of this technique. However, with modern CrA devices, these complications have become extremely rare. In a recent metanalysis evaluating CrA versus RFA for hepatic malignancies, the authors found that complications were slightly higher in the CrA group, but limited to thrombocytopenia and renal impairment [9]. In this regard, a heat-based track ablation device with new-gene+ration cryoprobes is available for a few months at our institution, which has been related with decreased bleeding complications [28].

One of the most important characteristics of CrA includes the intraprocedural monitoring of the necrosis area induced by the probe. In fact, the resulting iceball corresponding to the ablation area is well-defined and better evaluable in its extension when compared to RFA or MWA [4,6,13]. This allows for minimizing the risk of injuries to other tissues of organs. During percutaneous ablation, undesired collateral damages to structures near the ablation zone can be avoided by monitoring the temperature with one or more locally inserted thermocouples, and regulating the power of the probe accordingly. To note, this precautional use of thermocouples is not necessary with CrA, and they were not used in our experience.

Relative disadvantages of CrA, when compared to RFA and MWA, are the longer operative time and the higher costs. In our experience, the time required for the CrA procedure is longer than observed for heat-based techniques. However, the longer operative time might be counterbalanced by the note advantages of CrA, such as the ability to image the ice-ball formation and the almost complete absence of procedural pain. Even though it was not among the endpoints of our work, we have calculated a mean cost of approximately of EUR 4100 for every session of CrA, which is higher than observed for RFA and MWA (approximately EUR 3300 each), consistent with the literature. The issue of costs should be taken into account in radiological interventional centers involved in multidisciplinary treatment of liver malignancies.

One of the endpoints of the present work was complete ablation at 1 month, which was achieved in 92% of patients, in line with other reports. Interestingly, the rate of complete ablation was higher in metastases than HCC (*p* < 0.05). There is not a clear explanation for this result, and we speculated that the difference might be due to the small sample size in the two groups.

We have investigated the rates of local recurrences after CrA. The recurrence of disease in the liver remnant represents the main problem in the face not only of percutaneous CrA, but also of surgical resection and heat-based ablation methods used for HCC or liver metastases. In fact, there is a recognized high likelihood of recurrence of liver metastases from colorectal cancer even after resection (50–75% after first liver resection) [24]. As for HCC, nonlocal intrahepatic recurrence of HCC after RFA for early-stage HCC is reported to be 70–76% by 5 years [29]. It should be noted that the definitions of these endpoints are not uniform throughout the studies on CrA, thus rendering difficult the comparison of the outcomes. In our study, there was not a significant difference in the two groups regarding the rates of local recurrences in liver remnant, which occurred in 43% of HCC and 28% of metastases submitted to CrA during a follow-up of about 20 months (*p* = 0.289). These figures are consistent with other reports, although until now, there have been limited data on hepatic recurrence after successful CrA of HCC. In our previous report, when investigating the use of MWA in patients with HCC, the rate of local recurrence was 12.5% after a median follow up of 30 months [12]; however, a comparison between results of MWA and RFA at our center would be hampered by the small sample size at this moment. Ng et al., after a median follow up of 28 months, registered a recurrence in the liver remnant in 126 cases among 293 patients (43%) receiving CrA for colorectal metastases [14].

Local tumor progression ranges between 3% and 25% in the literature [15,17,19], and occurred in 9% of cases in our case series. In the experience from Littrup et al., including 212 patients with liver tumors treated with CrA, recurrences within the ice ablation zone, or within 1 cm of the ablation rim, were 5.5%, 11.1%, and 9.4% for HCC, colorectal, and non-colorectal metastases at a mean follow-up of 1.8 years, respectively [17]. In their metanalysis, Wu et al. found no difference in 6-month mortality and local tumor progression between patients who underwent CrA versus RFA for hepatic malignancies [9]. Similarly, for RFA and MWA, it should be taken into account that CrA can be redone [30,31,32], as we reported in the current series, whereas redo CrA was carried out for local tumor progression or local recurrence in 16% of cases.

The median survival of patients with HCC in the present series was 22 months. This finding compares favorably with Yang et al., who reported a median survival of 27 months among 74 patients treated with CrA for HCC abutting the diaphragm [33]. Interestingly, a prospective randomized trial including 360 patients showed an equal overall 5-year survival between patients who underwent percutaneous CrA or RFA for the treatment of HCC [34].

Although this aspect was not among the endpoints of our study, it is important to note that recent reports have focused on a possible effect of CrA ablation on immunomodulation other than simple tumor destruction [8,16,35]. In the era of immunotherapy, increasing interest has been directed towards the CrA-induced antitumor immune response, which may aid in tumor control and cure. The persistence of antigenic tissue in the liver after local ablation plays a central role. During the thawing phase of CrA, tumor cells within the iceball release into the blood stream intact tumor antigens, such as nuclear proteins, proinflammatory cytokines, and other molecules. These signals stimulate the natural immune response by causing the activation of macrophages, NK cells, granulocytes, cytokines, and dendritic cells. A combination of immunotherapy and CrA represents an interesting field of translational research, with promising applications in cancer management.

We acknowledge that this work has some limits, the main being its retrospective design and the small sample size. Furthermore, patients with both HCC and metastases from colorectal and non-colorectal origin were included, thus rendering difficult to evaluate long-term oncological results. Nonetheless, the series reported herein represents one of the few reports from a European institution investigating the use of CrA for liver malignancies.

## 5. Conclusions

In conclusion, different percutaneous ablation methods are available in patients with primary or secondary liver tumors not suitable for surgical treatment. However, no clear guidelines exist, thus the selection is largely dependent on availability of devices and operator’s preference. In particular, there is a relative scarcity of data regarding the using of CrA as an alternative to RFA or MWA. Our results confirm that CrA can be safely used in patients with liver tumors in close vicinity to critical structures, where the use of the latter two ablation techniques can be unadvisable. Further studies are needed to evaluate long-term oncological outcomes and to better define the role of CrA in the multidisciplinary approach to liver malignancies.

## Figures and Tables

**Figure 1 cancers-14-03018-f001:**
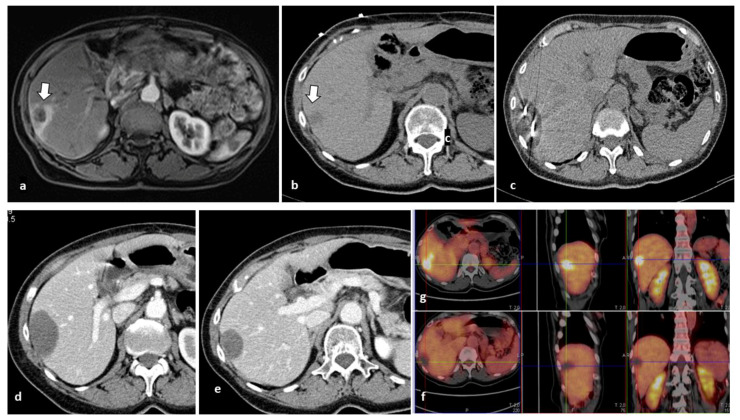
Complete tumor ablation of a single liver metastasis from colorectal cancer treated with CrA in a 79-year-old man. (**a**) Axial MRI and (**b**) axial CT scans show a 16-mm lesion in liver segment 8 (arrows), adjacent to the hepatic margin. (**c**) CT images obtained during CrA demonstrate placement of two *IceRod PLUS* cryoprobes, with the hypodense iceball encompassing the entire tumor mass. (**d**) CT obtained at 1-month follow-up shows the area of necrosis corresponding to the iceball, and no mass enhancement. (**e**) At the 6-month follow-up CT scan, a reduction in the size of the necrotic area without contrast enhancement was observed. (**f**) FDG-PET/CT scan obtained after 12 months CrA shows no FDG uptake, compared with baseline FDG/PET (**g**).

**Figure 2 cancers-14-03018-f002:**
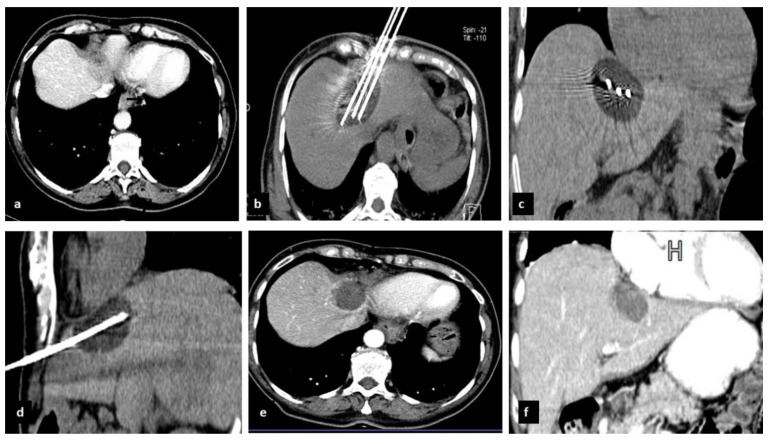
Complete CrA of a 3.5 × 2.2 mm ovarian metastasis located in segment 8 in a 62-year-old woman. (**a**) Axial CT contrast enhancement shows the lesion is located very close to the heart and superior vena cava (black arrows). (**b**) Axial CT scan performed during the freezing phase with three cryoprobes IceRod PLUS. (**c**) Coronal and (**d**) sagittal electronic reconstructions show the iceball very close to the pericardium. Axial (**e**) and coronal (**f**) contrast-enhanced CT scan after 1 month show a hypodense area with a rhyme of marginal enhancement due to granulation tissue reaction.

**Table 1 cancers-14-03018-t001:** Clinical and tumor characteristics of the study population.

Characteristic	Total Cohort (N = 49)	Patients with HCC (N = 11)	Patients with Liver Metastases (N = 38)
Gender			
Male	23 (47%)	9 (82%)	14 (37%)
Female	26 (53%)	2 (18%)	24 (63%)
Age (mean ± SD)	67.5 ± 10.4	67.8 ± 9.6	67.4 ± 10.8
Total no. of tumor lesions treated with CrA	64 (100%)	14 (22%)	50 (78%)
1 lesion	54 (84%)	12 (86%)	42 (84%)
2 lesions	5 (16%)	1 (14%)	4 (16%)
Tumor diameter, in mm (mean ± SD)	21.5 ± 10.7	19.0 ± 3.8	22.2 ± 11.9
Tumor location			
Right liver	24 (37.5%)	5 (36%)	19 (38%)
Left liver	40 (62.5%)	9 (64%)	31 (62%)

**Table 2 cancers-14-03018-t002:** Operative outcomes in the two study groups (HCC and metastases).

Outcome	HCC (14 Lesions in 11 Patients)	Metastases (50 Lesions in 38 Patients)	*p* Value
Technical success	14 (100%)	50 (100%)	-
Complete ablation at 1 month	11 (79%)	48 (96%)	0.032 *
Patients with local recurrence	5 (45%)	11(29%)	0.477
Total number of Local Recurrence	6 (43%)	14 (28%)	0.289
Patients with local tumor progression	1 (7%)	5 (10%)	0.105
Patients with distant metastases	1 (7%)	6 (12%)	0.607
Patients with minor complications	1 (9%)	5 (13%)	0.730
Redo liver CrA	2 (14%)	8 (16%)	0.864

* statistically significant.

## Data Availability

The data of the present study will be available under reasonable request.

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
