# Peer review of "The Increasing Role of CT-Guided Cryoablation for the Treatment of Liver Cancer: A Single-Center Report"

_cancers, 2022, doi:10.3390/cancers14123018_

Round 1

Reviewer 1 Report

The reviewer comments to the cancers-1714136

In this study, the authors aimed to evaluate the effectiveness and safety of CT-guided cryoablation (CrA) for the management of patients with liver tumors. Although the authors showed several data and images, the reviewer felt that these data were not sufficient to mention about the utility of CrA. The reviewer had several comments as follows.

Major comments:

  1. The data compared to heat-based method is not shown. Moreover, in the discussion chapter, the authors did not discuss about the effectiveness of CrA with referring data of heat-based devices. Therefore, the reviewer could not figure out the effectiveness of the CrA from this study.
  1. Regarding the local recurrence rate, 43% in HCC and 28% in metastases seem to be quite high compared to the data of heat-based devices. The reviewer think that the authors also may have many experiences of heat-based devices. Is the local recurrence rate of heat-based devices also high in your hospital?
  1. The time requiring for treatment is also important factor. if the authors have such data, please provide it. Moreover, please discuss about it with comparing the data of heat-based devices.

Minor comments:

  1. Although the figure legend of the Fig.2d mentioned head arrows, they are not indicated in the image. Please confirm it.
  1. In the result chapter (Page 6 line 179), range values of the tumor diameter are not indicated completely. The maximum value is not noted.
  1. The data of the Table 3 are duplication from Table 1 and 2. Please reconsider the configuration of tables.

Author Response

In this study, the authors aimed to evaluate the effectiveness and safety of CT-guided cryoablation (CrA) for the management of patients with liver tumors. Although the authors showed several data and images, the reviewer felt that these data were not sufficient to mention about the utility of CrA. The reviewer had several comments as follows.

Author: First of all, we are grateful to our reviewer for the effort in critically evaluate our work. We respect his/her opinion and hope that the new version, that was extensively modified and enriched according to the reviewers’ comments, will better define the role of Cryoablation in patients with inoperable liver cancer.  

Major comments:

  1. The data compared to heat-based method is not shown. Moreover, in the discussion chapter, the authors did not discuss about the effectiveness of CrA with referring data of heat-based devices. Therefore, the reviewer could not figure out the effectiveness of the CrA from this study.

The study was designed to evaluate the usefulness of CrA in patients with HCC and liver metastases; a comparation with percutaneous thermal techniques was not among the endpoints, and few studies compared cryo to heat-based methods. However, the following part was added in discussion, with the aim to better explain the role of percutaneous, image-guided ablation techniques in the treatment of HCC and met, and to highlight the proper indications and advantages of Cryo:  

The results of this study highlighted that CrA is an effective and safe method for the management of selected patients with liver tumors. Among locoregional therapies directed toward inducing selective tumor necrosis, percutaneous heat-based ablation techniques and cryoablation represent efficacious minimally methods that have been increasingly used in patients with liver cancer. Among them, RFA has been the most employed method, thus a robust evidence exists on the safety and efficacy of that technique in the treatment of metastases and HCC. Studies comparing RFA to resection in patients with HCC have shown that resection is generally associated with better survival outcomes        than RFA,, although surgery is associated with more complications and morbidity [18,20]. Nonetheless, some studies have underscored that RFA might be considered the first-line treatment in selected patients with HCC lesions ≤ 2 cm, provided that they are in an accessible location and away from major vascular and biliary structures and adjacent organs. Peng et al. reported a 5-year OS of 80% in patients with central HCC tumors ≤ 2 cm treated with RFA [21].

MWA has also been shown to be useful in selected patients with either unifocal or multifocal liver tumors [6,8,10-12], although data remain relatively limited. In the only randomized controlled study that compared RFA with percutaneous MWA, no significant differences were found between these two techniques in terms of therapeutic effectiveness, complications, and the rates of residual foci of untreated disease [22]. In         a randomized study that evaluating the efficacy of MWA versus surgical resection in the treatment of HCC, MWA was associated with lower disease-free survival rates with no differences in OS rates [23]. In a single institution report, we evaluated the use of MWA in 32 patients with 45 HCC tumors. Complete ablation was obtained in 94.1% of small tumors (< 3 cm). One-year OS was 82.7%, 2-year survival 68.9%, and 3-year survival 55.2% [12].

In general, MWA is associated to lower operative time than RFA, and also permit the simultaneous use of multiple antennas, thus resulting in a more versatile method. For these reasons, at our center MWA has largely replaced RFA in the percutaneous treatment of HCC and liver metastases, in line with current trends in many interventional radiology centers [24]. 

Moreover, the legend of Figure 1 highlights some advatanges of CrA

  1. Regarding the local recurrence rate, 43% in HCC and 28% in metastases seem to be quite high compared to the data of heat-based devices. The reviewer think that the authors also may have many experiences of heat-based devices. Is the local recurrence rate of heat-based devices also high in your hospital?

This is an interesting point. We have also reported the data from our experience in the use of MWA for HCC published in the Journal of Gastrointestinal Cancer (doi: 10.1007/s12029-017-9951-8.) The following part was added in discussion:

We have investigated the rates of local recurrences after CrA. The recurrence of disease in the liver remnant represents the main problem in the face not only of percutaneous CrA, but also of surgical resection and heat-based ablation methods used for HCC or liver metastases. In fact, there is a recognized high likelihood of recurrence liver metastases from colorectal cancer even after resection (50–75% after first liver resection) [24]. As for HCC, nonlocal intrahepatic recurrence of HCC after RFA for early-stage HCC is reported to be 70–76% by 5 years [29]. It should be noted that the definitions of these endpoints are not uniform throughout the studies on CrA, thus rendering difficult the comparison of the outcomes. In our study, there was not significant difference in the two groups regarding the rates of local recurrences in liver remnant, that occurred in 43% of HCC and 28% of metastases submitted to CrA during a follow-up of about 20 months (p=0.289). These figures are consistent to other reports, although up to now there are limited data on hepatic recurrence after successful CrA of HCC. In our previous report investigating the use of MWA in patients with HCC, the rates of local recurrence was 12.5% after a median follow up of 30 months [12]; however a comparison between results of MWA and RFA at our center would be hampered by the small sample size, at this moment. Ng et al, after a median follow up of 28 months, registered a recurrence in the liver remnant in 126 cases among 293 patients (43%) receiving CrA for colorectal metastases [14].

  1. The time requiring for treatment is also important factor. if the authors have such data, please provide it. Moreover, please discuss about it with comparing the data of heat-based devices.

We went back to patients’ files and reported the following in Results:

The following parts regarding operative time were added

in Results:

The mean duration of the CrA procedure was 67.3 ± 9.5 (range 55 to 77 minutes), that in our experience is longer than RFA (50-65 minutes) and MWA (45-60 minutes).

In discussion:

‘Relative disadvantages of CrA, when compared to RFA and MWA, are the longer operative time and the higher costs. In our experience, the time required for the CrA procedure is longer than observed for heat-based techniques. However, the longer operative time might be counterbalanced by the note advantages of CrA, such as the ability to image the ice-ball formation and the almost complete absence of procedural pain.’

Minor comments:

  1. Although the figure legend of the Fig.2d mentioned head arrows, they are not indicated in the image. Please confirm it.

To follow the suggestion of reviewer #4, who asked to remove some figures, in the revised version only the Figures 1 and 4 were left in the manuscript.

  1. In the result chapter (Page 6 line 179), range values of the tumor diameter are not indicated completely. The maximum value is not noted.

The sentence was corrected: “The mean tumor diameter was 2.15 cm (range 0.5 cm5 cm).”

  1. The data of the Table 3 are duplication from Table 1 and 2. Please reconsider the configuration of tables.

            We agree with our reviewer. We have eliminated Table 3 and modified Table 2 (now titled  “Tab. 2             Operative outcomes in the two study groups (HCC and metastases)”, by adding p-values (left       column) and by eliminating the column of ‘ total tumors’.

Reviewer 2 Report

Overall, this is a well-written paper on cryoablation for liver cancer (HCC and liver metastases)

It is a retrospective study not adding much new to our knowledge in this field.

Abstract: Well-written. You should add that the 92% tumor ablation rate was after 1 month

Materials and methods:

If CrA has been used for 10 years why did you only include the 6 years from 2015-2020?

The procedure is well described

Study endpoints:

When was the timing of the endpoints? The day after? 1 month? 1 year? Please specify

Table 3: you need to add units to tumor size (I presume mm)

Discussion: well-written

Author Response

REVIEWER #2

Overall, this is a well-written paper on cryoablation for liver cancer (HCC and liver metastases). It is a retrospective study not adding much new to our knowledge in this field.

We thank our reviewer for the comment. We would like to point out that few studies have addressed the outcomes of CrA in liver cancer, thus we believe that our paper, together with similar ones, can serve as a basis for further studies exploring the increasing role of thermal ablation in primary and metastatic liver tumors. Moreover, to the best of our knowledge, this is one of the few studies from an European institution regarding the use of CrA for liver cancer.

Abstract: Well-written. You should add that the 92% tumor ablation rate was after 1 month

This has been added in abstract.

Materials and methods:

If CrA has been used for 10 years why did you only include the 6 years from 2015-2020?

We stated that ‘In the last 10 years, RFA, CrA and MWA were used for thermal ablation of primary and metastatic malignancies…. We have started to use CrA in patients with liver cancer in 2015.’

The procedure is well described

Study endpoints:

When was the timing of the endpoints? The day after? 1 month? 1 year? Please specify

In Methods, we have stated that

-‘Technical success of CrA was considered when the tumor lesion was completely included in the iceball at the end of the procedure’.

- ‘Ablation was considered complete when no contrast enhancement was observed in arterial phase at 1-month CT-scan.’

- Local tumor progression was the increase in the diameters of the lesion. Steadiness or decrease in diameters of the ablation area was considered as successful ablation. Local recurrence was defined as the appearance of new tumor lesions in the remnant liver.

These two endpoints (LTP and LR) had not a “timing” but were registered as they appeared during follow-up.

Table 3: you need to add units to tumor size (I presume mm)

To respond to a request of reviewer #1, who asked for a rearrangement of the Tables, Table 3 was eliminated and results incorporated in Table 2.

Discussion: well-written

Reviewer 3 Report

The authors submitted a manuscript titled “The increasing role of CT-guided cryoablation for the treatment of liver cancer. A single-center report.” There is a total of 49 patients involved in this study, please do not prepare your article like a case report. The case reports are not considered by /Cancers/. Please well revise the manuscript. Despite this, the followings are some concerns and comments have been pointed out that the authors may want to consider.

  1. Line 41 Keywords: None of the keyword “Hepatocellular carcinoma” appears in the main context. I don’t think it is suitable to be a keyword.
  2. Line 43: The introduction section needs to be enriched. And please remove unnecessary self-citation.
  3. Line 77: The “3” should be superscript.
  4. Line 114, “−40 â—¦C” (superscript “â—¦”), line 115, “−40°C”: the format should be consistent. With or without a space between value and unit.
  5. Line 140: There is an extra space before the word “values”. Please check throughout the manuscript.
  6. Line 140: It should be “p<0.05” instead of just “<0.05”.
  7. Line 160: A space is needed after the word “margin”.
  8. Line 206: Please use italic p as it refers to a p-value. Please check throughout the manuscript.
  9. Line 209: Please use “Table” instead of “Tab.” Or defined it before using it. And please be homogenous throughout the manuscript.
  10. Line 209 Table 1: I’d suggest the authors just use “n = 49” that’s fine instead of “No. = 49”. Please consider and check throughout the manuscript.
  11. Line 213 Table 3: The meaning of the number in the bracket “()” should be mentioned.
  12. Line 283: It seems there is an extra “-“ following the word “colorectal”. Please double-check.
  13. Line 327: The “Please add:” should be deleted. Double-check.
  14. Line 335: There are 7 out of total of 31 references from the listed authors. This over 20% self-citation is too high.

Author Response

REVIEWER #3

The authors submitted a manuscript titled “The increasing role of CT-guided cryoablation for the treatment of liver cancer. A single-center report.” There is a total of 49 patients involved in this study, please do not prepare your article like a case report. The case reports are not considered by /Cancers/. Please well revise the manuscript. Despite this, the followings are some concerns and comments have been pointed out that the authors may want to consider.

We thank the reviewer for the comment. The study design was a monocentric case series. We have expanded the introduction and discussion in order to better respond to the requisites of Cancers for original articles. Although the number of patients was low, it reflects the sample size of other monocentric reports on the same topic.

  1. Line 41 Keywords: None of the keyword “Hepatocellular carcinoma” appears in the main context. I don’t think it is suitable to be a keyword.

The acronym HCC was used for hepatocellular carcinoma in the whole manuscript after first use. HCC was added to the keywords.  

  1. Line 43: The introduction section needs to be enriched. And please remove unnecessary self-citation.

We agree with our reviewer. The introduction has been rewritten to better focus on the background of current application of thermal ablation in liver malignancies, as well as to better define the aims of the work. Moreover, some epidemiological data regarding HCC and liver metastases have been added with relative references.Unnecessay sel-citations have been removed.

  1. Line 77: The “3” should be superscript.

This has been corrected. “…platelet count >100,000/mm3

  1. Line 114, “−40 â—¦C” (superscript “â—¦”), line 115, “−40°C”: the format should be consistent. With or without a space between value and unit.

Corrections have been made.

  1. Line 140: There is an extra space before the word “values”. Please check throughout the manuscript.
  2. Line 140: It should be “p<0.05” instead of just “<0.05”.

The beginning of the sentence has been corrected in: P-values < 0.05. The entire manuscript has been checked for extra spaces.

  1. Line 160: A space is needed after the word “margin”.
  1. The entire manuscript has been checked for extra spaces.

  1. Line 206: Please use italic p as it refers to a p-value. Please check throughout the manuscript.

Italic has been used for ‘p’ referring to a p-value throughout the manuscript. 

  1. Line 209: Please use “Table” instead of “Tab.” Or defined it before using it. And please be homogenous throughout the manuscript.

Corrected as requested.

  1. Line 209 Table 1: I’d suggest the authors just use “n = 49” that’s fine instead of “No. = 49”. Please consider and check throughout the manuscript.

Corrected as requested.

  1. Line 213 Table 3: The meaning of the number in the bracket “()” should be mentioned.

Table 3 has been eliminated as suggested from reviewer #2.

  1. Line 283: It seems there is an extra “-“ following the word “colorectal”. Please double-check.

Corrected as requested.

  1. Line 327: The “Please add:” should be deleted. Double-check.

Corrected as requested.

  1. Line 335: There are 7 out of total of 31 references from the listed authors. This over 20% self-citation is too high.

Reference list has been modified according to this suggestion, and new references have been added since introduction and discussion sections have been expanded. Please note that we have left only three self-citations [3,12,13] that we considered essential for the drafting of the manuscript, and also to report on the experience of our center in percutaneous ablation of tumors.

Reviewer 4 Report

  • This paper reports the experience of a single hospital, from an European country, on the use of cryoablation for the percutaneous treatment hepatic lesions (metastatic and hepatocarcinomas). The number of treated lesions is not large, as compared to other experiences already reported in literature, however the paper is, in my opinion, interesting. Overall, the results reported are in line with what already reported in literature, confirming usefulness of CrA in this field.

  • RFA, CrA and MWA were available in the authors ‘hospital in the study period (2015-2020). The number of RFA and MWA performed in the same period should be added.

  • “The decision to resort to CrA instead of heat-based methods was mainly conditioned by the proximity of the lesions to important anatomical structures (diaphragm, gallbladder, major biliary and vascular structures, chest wall, bowel and heart).”(page 2, lines 87-89). The concept is absolutely acceptable, however the lesions showed in figures 1, 2 and 3 could have been treated, without major problems, also with RFA or MWA. My suggestion is to remove figures 1, 2 and 3 and just leave figure 4.   Selection criteria to choose among the 3 techniques available should be, however, better defined. This, in my opinion, is a major point.

  • It should be interesting to add something on the costs of the procedures (as compared to RFA and MWA) and the post procedure length of stay in hospital.

  • “All CrA procedures have been performed under local anesthesia and conscious sedation (page 8, lined 235-236)”. This sentence should be added in the material and method paragraph.

  • Was an anesthesiologist present during the procedure or the radiologists performed conscious sedation?

  • Were the needles advanced under CT guidance? Or advanced under US guidance and the position controlled by CT?

  • Any differences in post-procedural pain in treated patients with subcapsular lesions as compared to RFA or MWA, this topic could be interesting also as possible selection criteria.

Author Response

REVIEWER #4

  • This paper reports the experience of a single hospital, from an European country, on the use of cryoablation for the percutaneous treatment hepatic lesions (metastatic and hepatocarcinomas). The number of treated lesions is not large, as compared to other experiences already reported in literature, however the paper is, in my opinion, interesting. Overall, the results reported are in line with what already reported in literature, confirming usefulness of CrA in this field.

We thank our reviewer for this comment. We have acknowledged that one of the limits of the study was the small sample size. Nonetheless, few monocentric experiences (especially from Europe) have been published on the use of CrA for the treatment of hepatic malignancies. As we stated also in reply to reviewer #1, we believe that our work may contribute to further large multicentric studies on the use of CrA and other percutaneous ablation techniques in hepatic cancers.

  • RFA, CrA and MWA were available in the authors ‘hospital in the study period (2015-2020). The number of RFA and MWA performed in the same period should be added.

In Methods we have added:

Between 2015 and 2020, a total of 1092 image-guided percutaneous ablation (including RFA, MWA, and CrA) were carried out for lung, breast, liver, kidney and osseous tumoral lesions. Sixty-four CrA, 15 RFA and 73 MWA were performed for liver malignancies.

  • “The decision to resort to CrA instead of heat-based methods was mainly conditioned by the proximity of the lesions to important anatomical structures (diaphragm, gallbladder, major biliary and vascular structures, chest wall, bowel and heart).”(page 2, lines 87-89). The concept is absolutely acceptable, however the lesions showed in figures 1, 2 and 3 could have been treated, without major problems, also with RFA or MWA. My suggestion is to remove figures 1, 2 and 3 and just leave figure 4.   Selection criteria to choose among the 3 techniques available should be, however, better defined. This, in my opinion, is a major point.

We are glad that the reviewer recognized the validity of our methods. Although we agree that cases depicted in figures 1-3 would have been treated also with MWA or RFA, we feel that at least another case (Figure 1) other than figure 4, might be left in the manuscript, also to respond to reviewer #3, who suggested to present the manuscript in a different form from a simple case report.

In the Methods the following part has been added to better define our policy to choose ablation techniques in patients with liver malignancies:

Patients with HCC or liver metastases measuring less than 3 cm in major diameter were usually treated with RFA, while those larger than 3 cm were primarily approached with MWA, although we have been gradually replaced RFA with MWA also for small tumor lesions. The decision to resort to CrA instead of heat-based methods was mainly conditioned by the proximity of the lesions to important anatomical structures (diaphragm, gallbladder, major biliary and vascular structures, chest wall, bowel, and heart).         

It should be interesting to add something on the costs of the procedures (as compared to RFA and MWA) and the post procedure length of stay in hospital.

We have added in discussion:

‘While it was not among the endpoints of our work, we have calculated a mean cost of approximately of 4100 Euros for every session of CrA, that is higher than observed for RFA and MWA (approximately 3300 Euros each), consistently with the literature. The issue of costs should be taken into account in radiological interventional centers involved in multidisciplinary treatment of liver malignancies.’

We have added in Results:

‘Usually, patients were discharged the day after the CrA procedure.'

“All CrA procedures have been performed under local anesthesia and conscious sedation (page 8, lined 235-236)”. This sentence should be added in the material and method paragraph.

The sentence has been added in methods.

Was an anesthesiologist present during the procedure or the radiologists performed conscious sedation?

This sentence was added in Methods:

An anesthesiologist performed the conscious sedation and monitored the patient during the procedure and the postoperative period.

Were the needles advanced under CT guidance? Or advanced under US guidance and the position controlled by CT?

As we have explained in Methods all the procedures were carried out under CT guidance.

Any differences in post-procedural pain in treated patients with subcapsular lesions as compared to RFA or MWA, this topic could be interesting also as possible selection criteria.

We have added in discussion:

It is recognized that the pain related to CrA is lower than other percutaneous ablation techniques [2,4,9,25-27]. We have observed this finding mostly in patients with tumor lesions in proximity to the Glisson’s capsule, where the level of sedation and use of opiates were lower than necessary in RFA or MWA.

Moreover, the legend of Figure 1 highlights some advatanges of CrA in pain control.

Round 2

Reviewer 1 Report

The authors well responded to reviewer's comments. The reviewer had no additional comment.

Reviewer 2 Report

The authors have responded well to the questions posed

Reviewer 3 Report

I do not have any further concerns now; except for double-checking to homogenous the format throughout the manuscript before publication.  Good luck.

Line 42: Please use italic p as it refers to a p-value. For example, line 43, line 44, line 231, line 233, line 234…, and so on.